# T-Edit: Triple-Branch Diffusion Anchoring for Consistent Editing

**Linsong Shan** [1]  **Laurence T. Yang** [1 2]  **Zecan Yang** [2 †]  **Shijie Lian** [1 3]  **Shijie Lv** [1]  **Qilin Yang** [1]

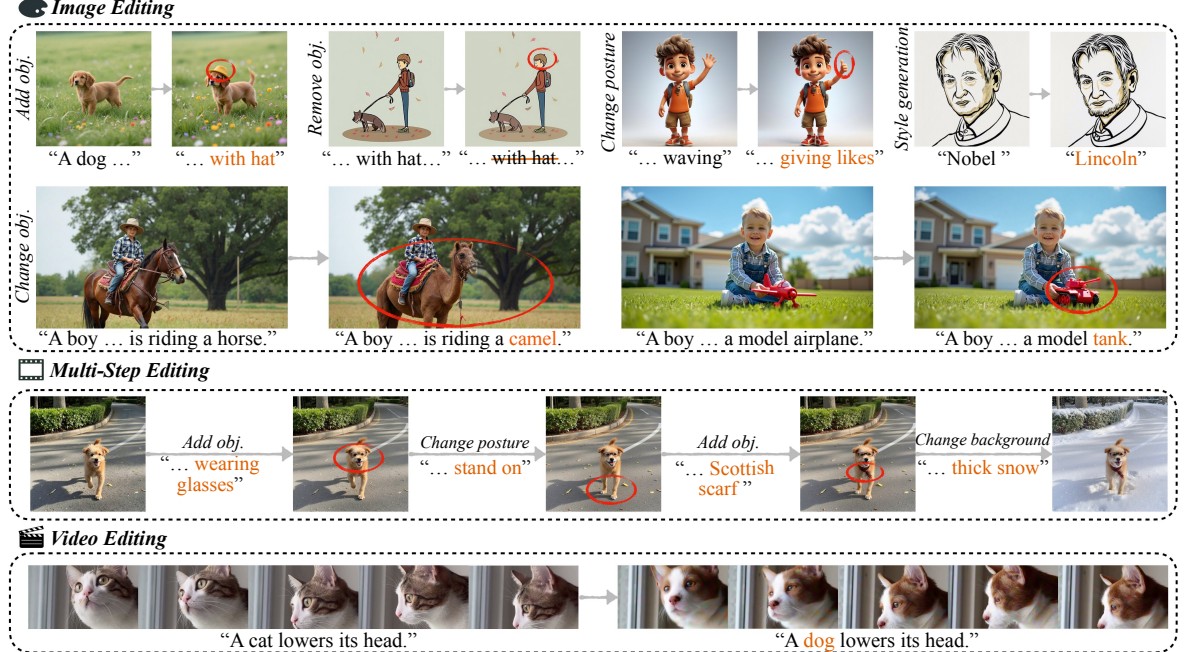

*Figure 1.* **Representative editing results produced by T-Edit.** Our train-free method enables high-fidelity editing that preserves structural consistency while precisely aligning with target prompts, in both single-step and multi-step scenarios.

## Abstract

While Multimodal Diffusion Transformers (MMDiTs) have achieved remarkable success in high-fidelity generation, maintaining semantic faithfulness and structural consistency during the image editing process remains a fundamental challenge. DiT-based editing is primarily limited by cumulative drift and semantic leakage induced by new textual conditions. To address these challenges, we propose T-Edit, a training-free framework that formalizes consistent editing as a trajectory anchoring process. T-Edit explicitly decouples the inversion, reconstruction, and editing trajectories, leveraging the reconstruction branch as a structural reference to achieve real-time compensation for deviations in the latent manifold. To further reveal the internal regulation mechanism of DiTs, we analyze the spatio-temporal heterogeneity of their layer-wise structural sensitivity and accordingly propose a Dynamic Vital Layer (DVL) localization mechanism based on information energy. Furthermore, addressing the asymmetry of textual perturbations in the frequency domain distribution, we introduce a frequency-aware strategy based on tensor Singular Value Decomposition (t-SVD) to anchor (TA) high-frequency structural components. Experiments show that T-Edit achieves state-of-the-art performance in both semantic alignment and structural fidelity, and can be seamlessly extended to multi-step editing and video scenarios, providing a new perspective for understanding and controlling the internal stability of DiTs. The code is available at GitHub.

†Corresponding author. [1]School of Computer Science and Technology, Huazhong University of Science and Technology [2]School of Computer Science and Artificial Intelligence, Zhengzhou University [3]Beijing Zhongguancun Academy. Correspondence to: Zecan Yang <zecanyang@gmail.com>.

*Proceedings of the $43^{rd}$ International Conference on Machine Learning*, Seoul, South Korea. PMLR 306, 2026. Copyright 2026 by the author(s).

# 1. Introduction

In recent years, text-to-Image (T2I) generation techniques have achieved remarkable progress. Early generation models based on U-Net (Ronneberger et al., 2015) backbones, such as Stable Diffusion (Rombach et al., 2022), pioneered high-quality synthesis but were inherently limited by the receptive field of convolutional networks in modeling long-range dependencies. To address this limitation, the new generation of MMDiTs-based models (Peebles & Xie, 2023), represented by Stable Diffusion 3 (Esser et al., 2024) and FLUX. 1 (Labs, 2024), leverages self-attention to capture global context, thereby producing more coherent scenes and finer structural details.

As T2I models mature, text-guided image editing has emerged as a critical downstream capability, supporting tasks such as object addition, replacement, background modification, and style transfer (Couairon et al., 2022; Shi et al., 2024; Alaluf et al., 2024; Xu et al., 2023; Huang et al., 2023). For example, as illustrated in Figure 1, a hat can be added to a dog, or a boy's gesture can be changed. Early editing methods, most notably Prompt-to-Prompt (Hertz et al., 2022), demonstrated that local edits could be achieved by manipulating attention maps in selected layers, inspiring a broad family of attention-based control techniques. Another dominant line of work relies on inversion-based editing, where an input image is mapped back to the diffusion latent space and regenerated with modified prompts, using techniques such as DDIM inversion (Song et al., 2020) and Null-Text Inversion (Mokady et al., 2023).

However, the transition to MMDiTs architectures fundamentally alters the failure modes of diffusion-based editing. Unlike U-Net backbones, where cross-attention layers act as a controlled gate between text and image. In MMDiTs, text and image representations are jointly updated across layers and timesteps, causing prompt perturbations to propagate globally through the denoising trajectory. As a result, even small semantic edits can trigger accumulated numerical drift, leading to background distortion, identity leakage, and structural collapse. This issue is particularly pronounced in multi-stage editing, where errors introduced at early steps compound over time and cannot be corrected by one-time inversion refinement or static attention manipulation.

To address the above limitations, we propose **T-Edit**, a novel training-free, three-branch editing paradigm that explicitly balances semantic accuracy and structural fidelity. Specifically, the inversion branch captures and hierarchically stores latent representations and key-value attention pairs at each step during DDIM inversion; the reconstruction branch reconstructs the original image from noise, progressively computing error signals between stored and reconstructed states to quantify editing drift; the editing branch injects new prompts while anchoring the diffusion process through stored information and reconstruction error corrections.

Furthermore, our analysis reveals two key observations underlying structural instability in MMDiTs editing. First, the layers that dominate structural preservation are neither fixed across samples nor consistent across denoising timesteps. Instead, structural sensitivity varies dynamically with image content and diffusion stage, making static layer heuristics inherently suboptimal. Second, prompt-induced perturbations manifest primarily as high-frequency deviations in attention representations, which progressively corrupt the fine-grained structure while propagating through the joint attention stream.

Motivated by these findings, T-Edit introduces two complementary mechanisms. **Dynamic vital-layer localization (DVL)** adaptively identifies the most structurally influential attention layers at each timestep, enabling targeted intervention without relying on fixed layer assumptions. Meanwhile, a **frequency-aware t-SVD Anchoring (TA)** strategy selectively suppresses high-frequency semantic perturbations while anchoring essential structural layouts within the spectral domain. Together, these mechanisms allow T-Edit to regulate semantic influence at its source, providing adaptive and step-wise structural guidance during diffusion.

In summary, our main contributions are as follows:

1. We propose T-Edit, a training-free framework for MMDiTs-based models, which explicitly decouples inversion, reconstruction-based error modeling, and prompt-driven editing to enable stable and controllable image manipulation.

2. By observing that (i) structurally vital attention layers vary across timesteps and samples, and (ii) editing-induced perturbations concentrate in high-frequency attention, we introduce **dynamic vital-layer localization** and **frequency-aware t-SVD Anchoring** for structural guidance during diffusion.

3. Extensive experiments demonstrate that T-Edit consistently achieves superior structural preservation and semantic accuracy in both single-step and multi-step editing, while remaining readily extensible across different MMDiTs-based architectures.

## 2. Related Work

### 2.1. Text-Guided Editing Methods

Text-guided image editing methods are primarily built upon pre-trained text-to-image (T2I) diffusion models. Early approaches relied on U-Net-based architectures, such as Stable Diffusion (Rombach et al., 2022), while recent advances have shifted toward MMDiTs-based designs with

joint text–image streams, fundamentally changing how semantic and structural information are entangled during denoising, leading to models such as Flux. 1 and SD 3.

Current editing methods can be broadly classified into training-based (Brooks et al., 2023; Li et al., 2024; Kawar et al., 2023; Ju et al., 2024) and training-free approaches (Tumanyan et al., 2023; Cao et al., 2023; Dong et al., 2023; Hertz et al., 2022; Meng et al., 2021; Zhu et al., 2025; Wang et al., 2024b; Avrahami et al., 2025). The former leverage annotated datasets and task-specific loss functions to achieve strong performance, while the latter are favored for their flexibility and efficiency.

Another key direction is inversion-based editing, which maps an image back to the diffusion latent space for controlled regeneration. DDIM Inversion is widely used, but its reliance on implicit reverse inference often leads to reconstruction errors—especially under Classifier-Free Guidance (Ho & Salimans, 2022), where conditioning intensifies inconsistencies and degrades editing quality.

To improve inversion accuracy and editing stability, various solutions have been proposed (Mokady et al., 2023; Dong et al., 2023; Miyake et al., 2025; Ju et al., 2023; Meiri et al., 2023; Cho et al., 2024; Wallace et al., 2023), including optimization-based methods, prompt tuning, and structural adjustments, providing more reliable reconstruction and control for diffusion-based editing.

## 2.2. Consistency in Diffusion Models

The core editing challenge is balancing semantic faithfulness and structural fidelity. This necessitates various consistency mechanisms. Inversion consistency, addressed by methods like Null-Text Inversion (NTI) (Mokady et al., 2023), focuses on accurate latent variable reconstruction. However, this one-time optimization fails to provide continuous structural correction during the generative process. Structural consistency aims for shape and background coherence. Techniques include masking or localized attention control (Huang et al., 2023) to protect specific regions. Relatedly, Feature Consistency methods, such as Stable Flow (Avrahami et al., 2025), attempt structural anchoring via fixed feature injection in Transformer layers. These solutions are often limited by their dependence on static masking or non-adaptive layer selection. Developing a real-time, adaptive, and training-free mechanism to handle structural drift remains the central difficulty.

## 3. Method

### 3.1. Challenges of Consistency

In text-guided image editing based on diffusion models (Ho et al., 2020), a common paradigm is to obtain a latent rep-

resentation through inversion and perform denoising generation with new text prompts. However, this process often introduces two intertwined sources of deviation, which significantly affect the structural and visual consistency between the edited image and the original one.

**Accumulated Structural Drift.** By viewing the generative process as a discretized solution to the ODE $dZ/dt = f(Z(t), t, c)$ (Song et al., 2020), the inversion stage aims to trace the trajectory backwards. However, the inherent approximation of $\epsilon_\theta$ and finite discretization steps introduce a persistent numerical residual. We define the accumulated structural drift as:

$$\delta_t^{drift} = Z_t - Z_t^R, \tag{1}$$

where $Z_t$ is the inverted latent and $Z_t^R = \Phi(Z_{t+1}^R, t, c)$ represents the reconstructed state from the ODE solver $\Phi$. While negligible in isolation, these discrepancies compound over successive timesteps, manifesting as geometric erosion even in a non-editing reconstruction.

**Architectural Semantic Bleeding.** Unlike the decoupled cross-attention paradigm in U-Net architectures, the unified joint-stream Transformer in DiT facilitates an all-to-all interaction between image and text tokens (Hu et al., 2025; Feng et al., 2025). This architectural shift allows the editing perturbation (formulated as $\delta_t^{edit} = \Phi(Z_t^R, t, c') - \Phi(Z_t^R, t, c)$) to penetrate irrelevant spatial regions, causing "semantic bleeding" that compromises global consistency.

Crucially, these two types of deviation are not independent. The pre-existing $\delta_t^{drift}$ destabilizes the latent manifold, making it increasingly susceptible to semantic perturbations. Conversely, the semantic perturbation amplifies the numerical instability of the ODE solver. Accurately modeling and mitigating these two types of intertwined errors is thus a key challenge for achieving consistent editing.

### 3.2. T-Edit: A Training-Free Three-Branch Paradigm

Taking FLUX. 1 as a representative implementation of MMDiTs, we adopt a training-free three-branch scheme to separate structural reference, reconstruction, and editing guidance (Figure 2 (a)).

We first invert the input image to obtain a latent reference trajectory $\{Z_t\}$ under the original prompt, which serves as a structural baseline. A parallel reconstruction pass using the same prompt yields a reconstructed trajectory $\{Z_t^R\}$, whose discrepancy from the reference reflects accumulated deviations introduced during inversion and denoising.

Editing is performed by running the diffusion process under the edited prompt $c'$, producing an editing trajectory $\{Z_t^E\}$. To prevent excessive structural drift, we incorporate

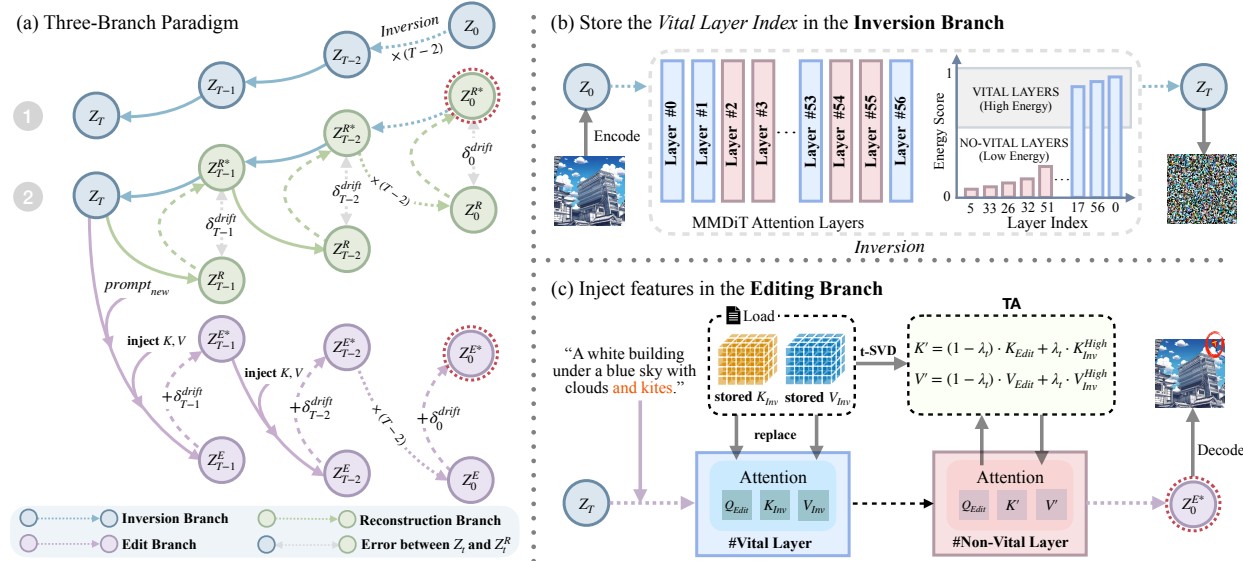

*Figure 2.* **T-Edit pipeline for image editing.** (a) illustrates the overall three-branch paradigm; (b) depicts the process of storing the vital layer indices $\{I_t\}$ during inversion; (c) shows how features are injected into the editing branch.

the observed discrepancy between the reference and reconstruction trajectories as a lightweight correction signal that aligns the editing trajectory with the structural baseline.

To ensure high-fidelity and non-destructive precise editing, the feature injection strategy must fundamentally address two pivotal questions within the MMDiTs layers: *Where to inject features* and *How to inject them*.

**Dynamic Vital Layer Localization.** Recent investigations have revealed a structural non-uniformity within the transformer hierarchy, where a small subset of so-called **vital layers** exert a disproportionate influence on geometric fidelity (Avrahami et al., 2025). Conventionally, these layers are identified via exhaustive, heuristic-based ablation and held static throughout the diffusion trajectory.

We perform the same ablation analysis under a diverse set of images, varying in scene type, resolution, seed, and denoising timesteps. In Figure 3, we visualize the ten layers with the lowest DINOv2 (Oquab et al., 2023) scores (i.e., the layers whose removal causes the largest structural change) using darker color tones. The results reveal substantial variability in the distribution of vital layers, indicating that fixed layer selections fail to generalize across different conditions.

Motivated by this observation, we seek a lightweight intrinsic signal that can identify vital layers dynamically during diffusion. In the self-attention of MMDiTs-based models, the Value tensor carries spatial features after mixing text & image tokens, and its variation is closely related to content structure. Let $V_t^{(l)}$ denote the Value tensor of the $l$-th layer at step $t$. We vectorize each $V_t^{(l)}$ and concatenate them

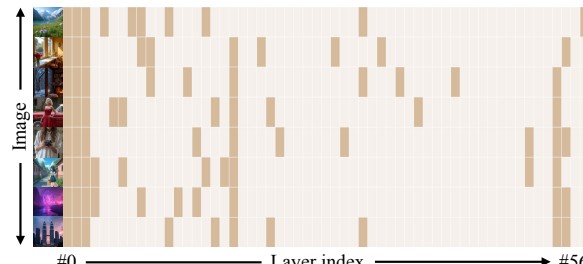

*Figure 3.* **Layer-wise ablation.** In each row, the dark blocks represent the vital layers selected based on DINOv2.

along the layer dimension to capture joint variation patterns:

$$V_t = \left[\text{vec}(V_t^{(1)}), \ldots, \text{vec}(V_t^{(L)})\right] \in \mathbb{R}^{d \times L}, \qquad (2)$$

where $d$ is the flattened feature dimension and $L$ is the number of MMDiTs layers. This matrix represents the joint variation space of all Value tensors at step $t$. We then apply Singular Value Decomposition (SVD) to $V_t$:

$$V_t = U_t \Sigma_t W_t^\top. \qquad (3)$$

Here, each row of $W_t \in \mathbb{R}^{L \times L}$ corresponds to the projection of a specific layer onto the global variation bases, and each singular value $\sigma_i$ reflects the energy of the $i$-th variation mode. Crucially, since SVD extracts the principal components of feature covariance, the dominant singular values correspond to global spatial correlations. Therefore, the term $\sigma_i |W_{t,l,i}|$ represents the contribution of the $l$-th layer to the $i$-th mode of high-energy structural variation.

Since $L \ll d$, the computational cost of this decomposition is small relative to the forward pass of the transformer. Based on this decomposition, we define the **Layer-wise Information Energy** as:

$$\mathcal{E}_t^{(l)} = \sum_{i=1}^{L} \sigma_i \cdot |W_{t,l,i}|. \tag{4}$$

Our experiment found that the $\Delta$ DINOv2 loss caused by removing that layer has a clear positive correlation with the corresponding normalized $\mathcal{E}_t^{(l)}$ (Figure 4), demonstrating that the information energy serves as a reliable proxy for the structural sensitivity of each layer.

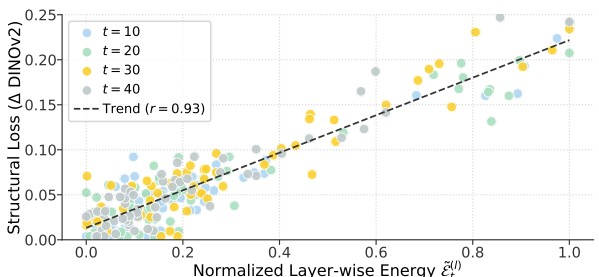

*Figure 4.* **Correlation analysis**. Validation of information energy and structural loss.

Instead of enforcing a rigid constraint on the number of layers, we introduce the Information Preservation Ratio $\tau$ as an adaptive structural bottleneck. Eq. 5 acts as a dynamic filter: it automatically expands the set $I_t$ during timesteps where structural signals are dispersed.

$$I_t = \left\{ l \ \middle| \ \sum_{l' \in I_t} \mathcal{E}_t^{(l')} \geq \tau \cdot \sum_{j=1}^{L} \mathcal{E}_t^{(j)} \right\}. \tag{5}$$

By sorting layers in descending order of $\mathcal{E}_t^{(l)}$ and accumulating their energies, we dynamically determine the number of vital layers at each timestep. In all experiments, we set $\tau = 0.9$, which consistently selects a small subset of high-energy layers due to the highly skewed energy distribution in MMDiTs. Figure 2(b) illustrates how the set $\mathcal{I}_t$ is recorded at each iteration.

**Frequency-aware t-SVD Anchoring.** To further investigate the changes induced by the new prompt's perturbation, we conducted a frequency-domain sensitivity analysis on the K/V weights. Specifically, we sought to determine how the perturbation is channeled within these non-vital layers by isolating changes into low-frequency (macroscopic structure) and high-frequency (local details) components. Experimental results (as shown in Figure 5) provide a key insight: across all diffusion timesteps, the average L2 norm

of the high-frequency component difference ($\Delta_{High}$) consistently and significantly exceeds that of the low-frequency component difference ($\Delta_{Low}$).

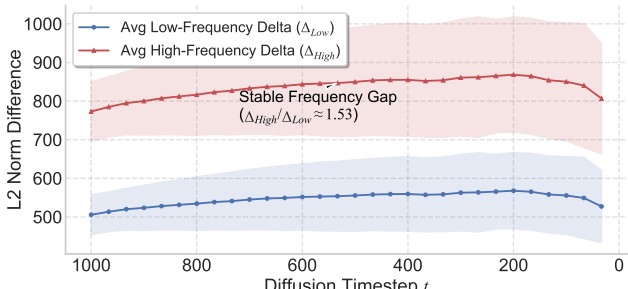

*Figure 5.* **Frequency-Domain Sensitivity to Prompt Change Over Time.**

This concentration of sensitivity in the high-frequency domain suggests that prompt-induced perturbations predominantly affect fine-grained details. Consequently, we propose a frequency-domain-based defense strategy: by injecting high-frequency signals into non-vital layers, mitigate the new prompt's excessive interference with the original details, thereby achieving more robust image protection.

Specifically, we design a novel attention mechanism based on t-SVD called **TA**. For each non-vital layer, TA applies t-SVD to the key-value tensors: tensors are transformed via FFT, decomposed using per-frequency SVD, and mapped back, and the resulting high-frequency components are injected into the editing branch.

To achieve a more refined and adaptive defense, we introduce a dynamic fusion factor $\lambda_t$, which is determined by the reconstruction error magnitude $||\delta_t^{drift}||^2$. We apply a sigmoid-based normalization to map the error magnitude to the range $[0, 1]$, yielding $\lambda_t = \sigma(||\delta_t^{drift}||^2)$. $\lambda_t$ increases when the reconstruction error is large, injecting a stronger structural protection signal into non-vital layers, and decreases when the error is small, allowing more flexibility for semantic editing.

We use this dynamic fusion factor $\lambda_t$ to fuse key-value pair, resulting in the guided key-value pairs $K'$ and $V'$:

$$\begin{aligned} K' &= (1 - \lambda_t) \cdot K_{\text{Edit}} + \lambda_t \cdot K_{Inv}^{\text{High}}, \\ V' &= (1 - \lambda_t) \cdot V_{\text{Edit}} + \lambda_t \cdot V_{Inv}^{\text{High}}, \end{aligned} \tag{6}$$

We construct a frequency-aware querying using the query $Q_{Edit}$ and the $(K', V')$. This modified attention:

$$\text{TA}(Q_{Edit}, K', V') = \text{softmax}\left( \frac{Q_{Edit} K'^{\top}}{\sqrt{d_k}} \right) V'. \tag{7}$$

Through this dynamic high-frequency guidance approach, the TA mechanism ensures that the defense signal can adjust

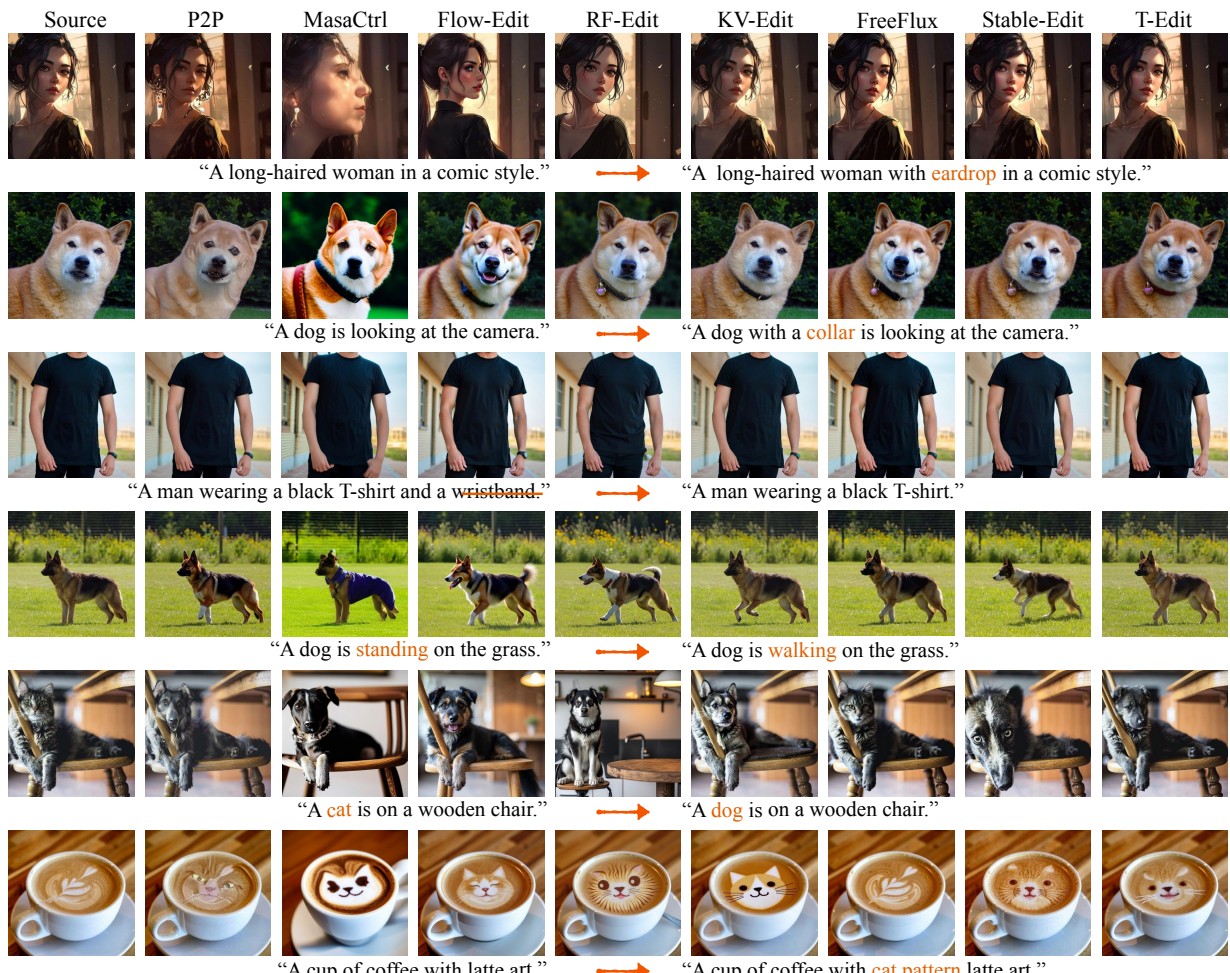

| Source | P2P | MasaCtrl | Flow-Edit | RF-Edit | KV-Edit | FreeFlux | Stable-Edit | T-Edit |

"A long-haired woman in a comic style." ⟶ "A long-haired woman with eardrop in a comic style."

"A dog is looking at the camera." ⟶ "A dog with a collar is looking at the camera."

"A man wearing a black T-shirt and a ~~wristband~~." ⟶ "A man wearing a black T-shirt."

"A dog is standing on the grass." ⟶ "A dog is walking on the grass."

"A cat is on a wooden chair." ⟶ "A dog is on a wooden chair."

"A cup of coffee with latte art." ⟶ "A cup of coffee with cat pattern latte art."

*Figure 6.* **Qualitative comparison of image editing.** Compared with the classic U-Net-based methods and the most advanced MMDiTs-based methods, our method demonstrates superior semantic loyalty and structure preservation. **Best viewed with zoom in.**

its strength in real-time according to the image's protection needs ($\delta_t^{drift}$), thereby adaptively preserving fine-grained details while maintaining editing flexibility.

# 4. Experiments

## 4.1. Setup

**Compared Methods.** To assess the effectiveness of our method, we compare it with seven representative approaches in the field of image editing. Specifically, Prompt-to-Prompt (P2P) (Hertz et al., 2022), and MasaCtrl (Cao et al., 2023) are built upon the U-Net framework, while Flow-Edit (Kulikov et al., 2025), RF-Edit (Wang et al., 2024b), KV-Edit (Zhu et al., 2025), FreeFlux (Wei et al., 2025), and Stable Flow (Avrahami et al., 2025) based on the DiT framework. Among them, KV-Edit introduces a mask-guided strategy to improve local editing precision. In general, these methods reflect a broad spectrum of the editing paradigms.

**Datasets and Evaluation Metrics.** Experiments are conducted on PIE-Bench (Ju et al., 2023). We evaluate the performance across three key dimensions: visual fidelity, semantic alignment, and structural preservation.

The FID (Heusel et al., 2017) measures distributional similarity between generated and real images, indicating image realism. For semantic alignment, we use the CLIP Score (Radford et al., 2021), which calculates cosine similarity between embeddings from a vision-language model. LPIPS (Zhang et al., 2018) and PSNR (Wang et al., 2004) jointly evaluate structural consistency, assessing perceptual differences and pixel-level fidelity, respectively.

## 4.2. Image Editing

**Quantitative Comparison.** Table 1 presents the quantitative results on PIE-Bench. In terms of generation quality, our method achieves the lowest FID value. Simultaneously, the highest CLIP score demonstrates that the images generated

by T-Edit exhibit exceptional semantic fidelity to the target text. Crucially, regarding metrics measuring structural consistency (PSNR and LPIPS), T-Edit achieves performance comparable to the mask-based method KV-Edit. This indicates that even when compared against methods specialized in structural preservation, our approach effectively maintains the structural and background consistency.

*Table 1.* **Quantitative comparison of image editing.** Quantitative comparison across four metrics on PIE-Bench.

| Method | FID ↓ | CLIP ↑ | LPIPS ↓ | PSNR ↑ |
|---|---|---|---|---|
| P2P (ICCV'23) | 131.55 | 28.95 | 0.4732 | 14.98 |
| MasaCtrl (ICCV'23) | 122.55 | 30.68 | 0.5002 | 13.60 |
| Flow-Edit (ICCV'25) | 77.65 | 30.33 | 0.2627 | 18.15 |
| RF-Edit (ICML'25) | 65.16 | 30.27 | 0.3677 | 19.82 |
| KV-Edit (ICCV'25) | 102.42 | 29.10 | **0.0950** | **27.17** |
| FreeFlux (ICCV'25) | 101.57 | 29.33 | 0.1865 | 22.42 |
| Stable Flow (CVPR'25) | 87.13 | 29.14 | 0.1527 | 20.72 |
| T-Edit (ours) | **62.22** | **31.55** | 0.1088 | 23.89 |

**Qualitative Comparison.** Figure 6 provides a visual comparison of different editing methods. P2P and MasaCtrl, both based on U-Net architectures, introduce modifications to the original image that exceed the specifications of the target text. Due to its non-inversion nature, Flow-Edit produces images exhibiting significant divergence from the source image. When executing *Change* tasks (e.g., modifying a dog's action), RF-Edit compromises the original structure and introduces undesired artifacts. KV-Edit achieves the highest structural consistency by strictly confining edits to masked regions, but this approach fails in *Remove* tasks. FreeFlux performs poorly in non-rigid changes. Stable Flow fails to maintain the balance between semantic fidelity and structural preservation when processing object replacement tasks, resulting in reduced structural consistency between edited and original images.

In contrast, our T-Edit strikes an excellent balance between semantic fidelity and structural consistency. Both quantitative and qualitative experiments demonstrate that T-Edit holds a distinct advantage in overall performance.

### 4.3. Multi-step Editing

Furthermore, we conduct comparative experiments on multistep sequential image editing to further validate the structural preservation advantages of our proposed method. Given the suboptimal performance of U-Net-based methods in prior experiments, we restrict comparisons to MMDiTs-based editing approaches. Qualitative results visualized in Figure 7 demonstrate that KV-Edit and our T-Edit method outperform counterparts in maintaining semantic alignment and structural consistency. However, KV-Edit encounters challenges in color modifications, while Stable Flow exhibits severe visual drift during the second editing step. Quantitative analysis in Table 2 reveals that mask-guided

KV-Edit achieves optimal structural metrics (LPIPS) due to region-constrained editing, yet its lower CLIP scores indicate compromised faithfulness to target text. In summary, across three editing steps, the proposed T-Edit maintains high textual faithfulness while closely approaching the structural consistency of mask-based KV-Edit.

*Table 2.* **Quantitative comparison of Multi-step editing.** CLIP measures semantic alignment, LPIPS measures structural retention.

| Method | CLIP ↑ | | | LPIPS ↓ | | |
|---|---|---|---|---|---|---|
| | Step1 | Step2 | Step3 | Step1 | Step2 | Step3 |
| Flow-Edit | 31.36 | 29.56 | 29.01 | 0.3031 | 0.3384 | 0.5316 |
| RF-Edit | 32.18 | 29.15 | 27.02 | 0.2964 | 0.3507 | 0.4264 |
| KV-Edit | **33.38** | 28.68 | 29.90 | **0.0819** | **0.1133** | **0.2618** |
| FreeFlux | 32.35 | 30.85 | 28.34 | 0.2684 | 0.3568 | 0.4235 |
| Stable Flow | 32.26 | 30.67 | 29.45 | 0.2922 | 0.5289 | 0.5849 |
| T-Edit (ours) | 33.12 | **32.91** | **31.09** | 0.1122 | 0.1984 | 0.3502 |

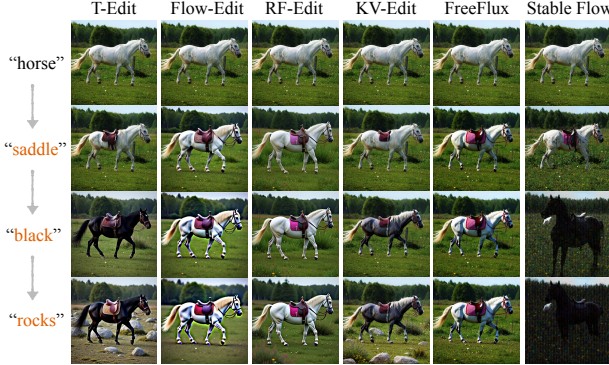

*Figure 7.* **Qualitative comparison of Multi-step editing.** Perform operations as adding a saddle, changing the color, and altering the background on the white horse in sequence.

### 4.4. Extension to Video MMDiTs Models

Video generation is an extension of image generation along the temporal dimension. To validate the generality and robustness of our T-Edit, we further applied it to the video editing task. The HunyuanVideo model (Kong et al., 2024) was used as the base, which is also built on the DiT architecture and integrates both double-flow and single-flow blocks. This structure is similar to the FLUX. 1, ensuring that T-Edit can be naturally migrated. To ensure temporal coherence and mitigate potential inter-frame flickering induced by dynamic layer transitions, we implement a first-frame anchoring strategy, where the vital layer indices $\{I_t\}$ for the entire sequence are determined solely based on the spectral analysis of the initial frame. Experiments are conducted on the Ditto-1M dataset (Bai et al., 2025), which contains diverse video clips with 36–72 frames at a resolution of 832×480 pixels. We compare T-Edit against three representative video editing methods: FateZero (Qi et al., 2023), COVE (Wang et al., 2024a) and RF-Edit.

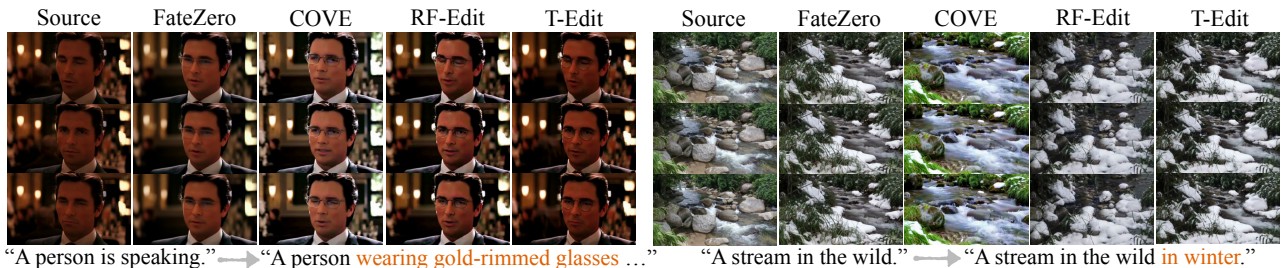

*Figure 8.* **Qualitative comparison of Video editing.** T-Edit maintains frame-to-frame structural consistency.

*Table 3.* **Quantitative evaluation of video editing.** Temporal LPIPS measures frame-to-frame perceptual consistency, and Flow error measures optical flow-based warping error.

| Method | FID ↓ | CLIP ↑ | Temporal LPIPS ↓ | Flow Error ↓ |
|---|---|---|---|---|
| FateZero | 85.12 | 32.45 | 0.210 | 0.145 |
| COVE | 92.50 | 30.12 | 0.245 | 0.162 |
| RF-Edit | 68.45 | 32.10 | 0.198 | 0.138 |
| T-Edit (Ours) | **65.87** | **33.28** | **0.182** | **0.131** |

*Table 4.* **Quantitative ablation of key components.** Different model variants are evaluated under identical settings.

| Configuration | FID ↓ | CLIP ↑ | LPIPS ↓ | PSNR ↑ |
|---|---|---|---|---|
| Base | 65.52 | 31.12 | 0.2270 | 18.54 |
| + Error Correction | 65.31 | 30.75 | 0.1540 | 19.22 |
| + Fixed VL | 64.12 | 31.32 | 0.1325 | 20.95 |
| + Dynamic VL | 64.15 | 31.50 | 0.1237 | 22.88 |
| Full T-Edit | **62.22** | **31.55** | **0.1088** | **23.89** |

The quantitative results in Table 3 indicate that T-Edit achieves the lowest FID, highest CLIP score, and lowest temporal LPIPS and flow error, indicating competitive visual quality, semantic alignment, and temporal consistency. Figure 8 illustrates representative editing results, including object replacement and global scene changes. Compared with baseline methods, which often alter or lose original structures, T-Edit generally preserves key structural elements across frames. These results suggest that the proposed approach can provide effective structural anchoring while maintaining coherent edits throughout the video.

### 4.5. Ablation Study

We conduct rigorous ablations to analyze each component's contribution and validate design choices motivated by MMDiT's structural characteristics.

Starting from direct generation using the target prompt (**Base**), we progressively incorporate: (i) reconstruction-based error correction, (ii) feature injection at fixed vital layers (corresponding to the assumption adopted in Stable Flow), (iii) dynamic vital layer localization, and (iv) frequency-aware injection via the TA module in non-vital layers. Quantitative results are reported in Table 4.

Comparing **Base** with **+ Error Correction**, we observe a substantial reduction in LPIPS, confirming inversion drift is a source of structural degradation in MMDiTs editing. The limited improvement in PSNR and the slight decrease in CLIP score indicate error correction alone is insufficient to control semantic leakage without explicit feature regulation.

Introducing feature injection at fixed vital layers (**+ Fixed VL**) yields clear gains in structural fidelity, as reflected by improved PSNR and LPIPS. Replacing the fixed layer configuration with dynamically localized vital layers (**+ Dynamic VL**) results in a further ∼2 dB increase in PSNR and consistently improves LPIPS, indicating that vital layers in MMDiTs editing are content- and prompt-dependent and cannot be adequately captured by a static layer set.

Finally, integrating the TA module into non-vital layers (**Full T-Edit**) leads to consistent improvements in structural metrics while preserving semantic alignment. This demonstrates that frequency-aware injection complements dynamic vital layer control by suppressing prompt-induced global structural corruption in self attention.

Overall, the ablation results show that robust editing requires the combined effect of explicit error correction, dynamic vital layer localization, and frequency-aware feature injection, each addressing a distinct failure mode inherent to MMDiTs architectures, and their cooperation is essential for achieving robust structural fidelity and semantic alignment.

## 5. Conclusion

This work investigates the non-uniformity of attention layers and the asymmetric evolution patterns of prompt perturbations in MMDiTs when subjected to semantic interventions. The introduction of DVL and TA mechanisms elucidates the intrinsic steady-state logic for maintaining trajectory consistency. Experimental results validate the generalizability of T-Edit across multiple models and provide a reference path for understanding the evolution of latent space manifolds in diffusion models from a fundamental perspective.

## Acknowledgements

This work was supported by the National Natural Science Foundation of China under Grant U23A20300.

## Impact Statement

This work improves the fidelity and controllability of image editing. While useful for creative applications, it may also make manipulated images harder to detect.

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

# A. Implementation and Module Design Details

## A.1. Temporal Scheduling and Validity of Early-Stage Intervention

The analysis in the main paper focuses on operations within individual denoising steps. Beyond this, we investigate how the temporal range of feature injection influences the balance between structural preservation and semantic flexibility. Specifically, we conduct a controlled study where feature injection is applied during the first $t = 1, 3, 5, 10, 20$ denoising steps, while the remaining steps proceed without intervention.

As shown in Figure 9, injecting features within the first 10 steps achieves a favorable trade-off, preserving structural details while maintaining strong semantic alignment with the target prompt. When the injection range becomes excessively long, the guidance overly constrains the generation trajectory, suppressing semantic adaptation to the target prompt. We refer to this effect as *structural overfitting*. These results highlight the importance of temporally limited intervention in diffusion editing, particularly for joint-stream MMDiTs models.

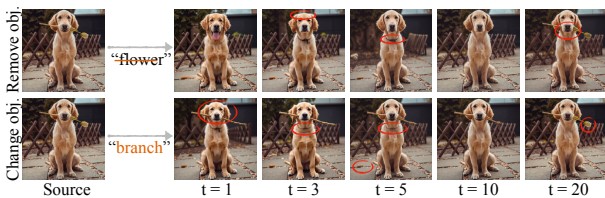

*Figure 9.* **Analysis of the injection timestep.** The t represents injection in the first t timesteps.

To further validate the design choice of early-stage error correction, we examine whether numerical drift in early denoising steps is sensitive to the specific text prompt. Our reconstruction-based correction implicitly assumes that early-stage drift is largely prompt-agnostic, allowing error vectors computed under the source prompt to be applied during editing.

Concretely, we first run the standard reconstruction branch using the source prompt $c$ and record the error vector at each denoising step,

$$\vec{e}_{src}(t) = Z_t - Z_t^R(c).$$

We then perform a counterfactual reconstruction using the target prompt $c'$, while forcibly resetting the latent variable to the same inversion trajectory $Z_t^{inv}$ at each step to control for accumulated deviations, yielding

$$\vec{e}_{tgt}(t) = Z_t - Z_t^R(c').$$

The cosine similarity between $\vec{e}_{src}(t)$ and $\vec{e}_{tgt}(t)$ is computed at each denoising step.

As shown in Figure 10, the cosine similarity consistently exceeds 0.8 during the first ten denoising steps, indicating strong alignment between error directions under different prompts. The similarity temporarily decreases around step 14 as prompt-specific semantic guidance becomes dominant, before increasing again in later steps as the sampling process converges. This observation supports our choice of restricting error correction to early denoising stages, where numerical drift is largely prompt-agnostic and can be reliably compensated using source-based reconstruction.

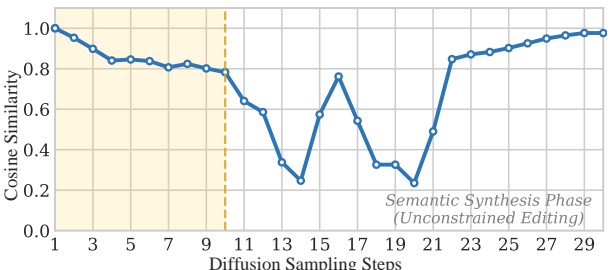

*Figure 10.* **Prompt-agnosticity of early-stage drift.** Cosine similarity between error vectors computed under the source and target prompts across denoising steps.

## A.2. Frequency Decomposition Strategies

For non-vital layers, T-Edit applies frequency-aware modulation rather than direct feature replacement. This modulation relies on tensor singular value decomposition (t-SVD) to separate attention representations into structure-dominant and detail-sensitive components.

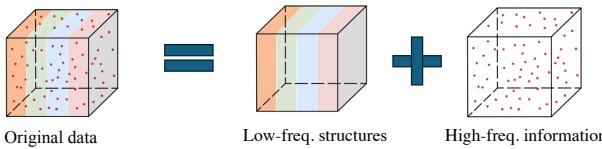

*Figure 11.* **Illustration of t-SVD.** Low-frequency structures and high-frequency information are separated from the original data.

Compared to channel-wise Fourier transforms or spatial spectral filtering, t-SVD operates directly on tensor representations and avoids imposing fixed frequency bases that may not align with learned attention patterns. In practice, we apply t-SVD to the key and value tensors of non-vital layers with shape $H \times L \times D$, performing FFT along the channel dimension and then per-frequency low-rank SVD on each frequency slice. The resulting high-frequency components are fused into the editing branch, with their contribution modulated by the dynamic fusion factor $\lambda_t$.

From an efficiency perspective, t-SVD is applied only to non-vital layers and only during early denoising stages. Moreover, the decomposition is performed during the inversion stage and reused during editing, introducing no additional computation in the forward denoising process.

### A.3. Fixed vs. Adaptive Injection Strength

We compare fixed-$\lambda$ strategies with the adaptive modulation used in T-Edit, where the effective injection strength is dynamically coupled with reconstruction error magnitude. Under fixed $\lambda$, strong modulation can overly suppress semantic adaptation in later steps, while weak modulation fails to correct early numerical drift, leading to accumulated structural deviation.

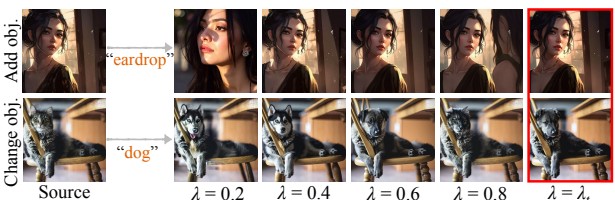

*Figure 12.* **Fixed vs. adaptive high-frequency injection.** The adaptive $\lambda_t$ dynamically balances structural correction and semantic flexibility across denoising stages.

### A.4. Editing Workflow under the Three-Branch Paradigm

T-Edit follows a training-free editing workflow built upon a three-branch paradigm, consisting of an inversion branch, a reconstruction branch, and an editing branch. The inversion branch records structure-related information from the input image, while the reconstruction and editing branches are jointly executed during generation to provide online structural anchoring and semantic modification, respectively.

**Inversion Branch.** Given an input image $X$, we first encode it into the latent space as $Z_0$ and apply DDIM inversion to obtain the terminal noise latent $Z_T$. During inversion, we record the intermediate latent states $\{Z_t\}$ as well as the key-value tensors from all self-attention layers. To identify structurally vital layers at each denoising step, we apply **Dynamic Vital-Layer Localization** to adaptively identify the vital layers set $I_t$. These recorded features are later used for both vital-layer replacement and frequency-aware modulation during editing.

**Synchronized Reconstruction and Editing.** Starting from the shared noise latent $Z_T$, we perform synchronized denoising using two parallel branches. The reconstruction branch is conditioned on the source prompt $c$ and serves as a structural reference, while the editing branch is conditioned on the target prompt $c'$. At each denoising step, attention features are injected into the editing branch prior to the forward update. Specifically, for layers in $I_t$, we directly replace the attention key-value pairs with those recorded during inversion. For all remaining layers, we apply frequency-aware modulation via the TA module, which selectively injects high-frequency components to suppress semantic leakage

while preserving structural integrity.

**Online Error Correction.** After each denoising step, we compute the deviation between the reconstructed latent and the inversion reference at the same timestep and use this deviation to correct the edited latent. This online correction suppresses accumulated numerical drift while maintaining sufficient flexibility for semantic editing.

Algorithm 1 summarizes the overall workflow of the inversion and synchronized generation process.

---

**Algorithm 1** Overview of T-Edit Editing Workflow

---

**Input**: Image $X$, source prompt $c$, target prompt $c'$
**Output**: Reconstructed image $\hat{X}^R$, edited image $\hat{X}^E$

1: Encode $X$ to latent $Z_0$ and perform DDIM inversion to obtain $Z_T$
2: **for** $t = 0$ to $T$ **do**
3:     Record latent $Z_t$ and attention features $\{K_t, V_t\}$
4:     Identify vital layers $I_t$ via SVD on $\{V_t^{(l)}\}$
5: **end for**
6: Initialize reconstruction and editing branches from $Z_T$
7: **for** $t = T$ to $0$ **do**
8:     Inject recorded features into editing branch (vital-layer replacement + TA)
9:     Perform synchronized denoising for reconstruction and editing
10:     Apply error correction to edited latent
11: **end for**
12: Decode final latents to obtain $\hat{X}^R$ and $\hat{X}^E$

---

### A.5. Vital Layer Selection: Additional Analysis

In the main paper, we discussed the limitations of prior methods that adopt fixed vital layer sets for all images. Notably, Stable Flow identifies vital layers by performing layer-wise ablation on a batch of 64 images, averaging the results to select a fixed set of layers for all editing tasks. However, we argue that this strategy lacks generality, as it assumes a universal importance distribution across all samples.

To support this claim, we conducted a controlled experiment on eight randomly selected images. For each image, we applied the same ablation procedure as in Stable Flow—removing the features at each attention layer independently and measuring the corresponding reconstruction error. The specific results are shown in Figure. 13.

## B. Runtime and Memory Analysis

This section reports the runtime and memory characteristics of T-Edit during inference, and compares them with representative editing baselines.

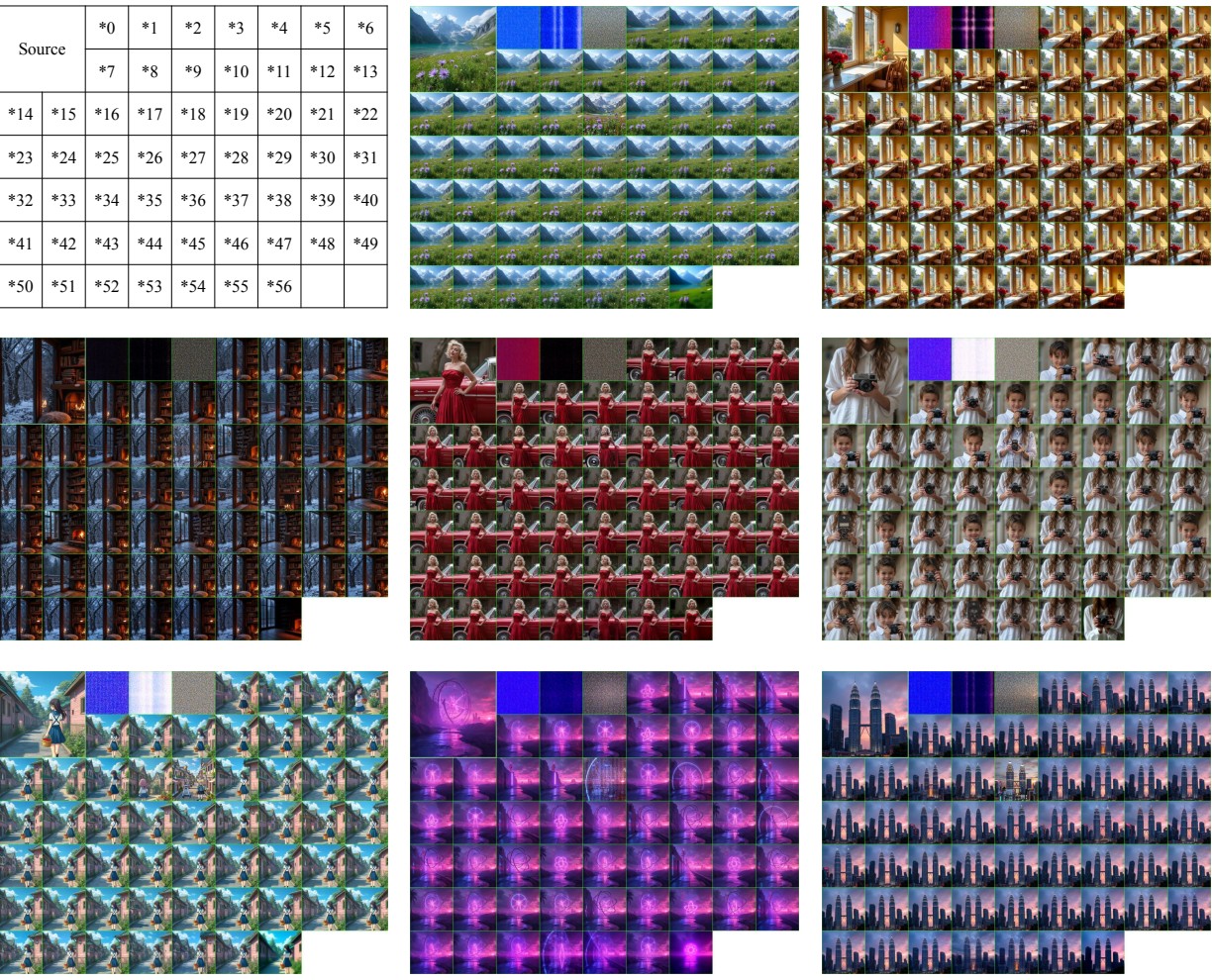

*Figure 13.* **Visualization of layer-wise ablation.** $*x$ represents the result of skipping the $x$-th layer.

## B.1. Inference Time

Table 5 reports the average inference time per image under identical hardware and resolution settings. The runtime of T-Edit is dominated by two components: the DDIM inversion process and the synchronized reconstruction–editing generation, both of which are standard operations in inversion-based editing frameworks.

Optimization-based methods such as RF-Edit exhibit substantially higher latency due to iterative optimization loops. Flow-Edit avoids inversion and thus shows lower runtime, but offers limited control in structure-sensitive editing scenarios. T-Edit adopts random SVD for time optimization, achieving a total runtime comparable to other inversion-based methods such as KV-Edit and FreeFlux.

The dynamic vital layer localization involves analysis of the high-dimensional Value tensor matrix $V_t \in \mathbb{R}^{d \times L}$. Given the architecture of MMDiTs, the feature dimension $d$ is typically several orders of magnitude larger than the number

*Table 5.* **Average total inference time per image.** All methods are evaluated on a NVIDIA A100 GPU under 30 diffusion steps.

| Method | Inversion | Total Runtime (s) ↓ | |
|---|---|---|---|
| | | $512 \times 512$ | $1024 \times 1024$ |
| P2P (Hertz et al., 2022) | ✓ | 83.32 | 245.15 |
| MasaCtrl (Cao et al., 2023) | ✓ | 22.31 | 68.45 |
| Flow-Edit (Kulikov et al., 2025) | ✗ | 30.33 | 82.10 |
| RF-Edit (Wang et al., 2024b) | ✓ | 65.57 | 126.36 |
| KV-Edit (Zhu et al., 2025) | ✓ | 39.39 | 112.54 |
| FreeFlux (Wei et al., 2025) | ✓ | 54.53 | 128.10 |
| Stable Flow (Avrahami et al., 2025) | ✓ | 31.96 | 94.22 |
| T-Edit (Ours) | ✓ | 43.37 | 104.45 |

of layers $L$, and $d$ scales quadratically with image resolution. To ensure the scalability of T-Edit, we utilize the Method of Snapshots by performing eigendecomposition on the compact $L \times L$ covariance matrix $C = V_t^\top V_t$:

$$V_t^\top V_t = W_t \Sigma^2 W_t^\top. \tag{8}$$

By shifting the computation from the resolution-dependent dimension $d$ to the constant layer dimension $L$. As shown

in Table 5, this optimization allows T-Edit to maintain competitive inference speed even at $1024 \times 1024$ resolution.

## B.2. Memory Usage

During inference, the GPU memory footprint of T-Edit is primarily determined by the underlying MMDiTs-based diffusion backbone (e.g., FLUX. 1). The additional memory overhead mainly arises from the Reconstruction Branch, and its GPU usage is comparable to that of other attention-based editing methods (e.g., MasaCtrl and Stable Flow) that inject attention features from an auxiliary batch guided by the original text prompt. Intermediate features, including latent states, dynamically selected vital layer indices, and attention key-value tensors, are cached on CPU and accessed sequentially during editing, so they do not increase GPU memory usage during the denoising process. Overall, T-Edit maintains GPU memory requirements comparable to existing methods.

## C. Additional Visualization Results

This section provides additional qualitative results that further illustrate the editing behavior of T-Edit across diverse scenarios. All samples are generated using the same configuration as in the main paper.

### C.1. Single-Step Editing with Baseline Comparison

Figure 14 presents more single-step editing examples, along with comparisons to representative baselines. These examples further highlight the structural consistency and semantic controllability of T-Edit under a variety of prompts.

### C.2. Multi-Step Editing Scenarios

Figure 15 shows representative multi-step editing results produced by T-Edit, where sequential modifications are applied to the same input. These visualizations demonstrate that the method remains stable across compositional prompts and preserves consistency throughout the editing trajectory.

### C.3. Extension to Stable Diffusion 3

To verify the architectural universality of T-Edit, we extended our evaluation to Stable Diffusion 3. SD3 similarly features an MMDiTs structure, where text and image tokens interact within a joint flow. As shown in Figure 16, T-Edit achieves high-fidelity attribute editing and local modifications on SD3, maintaining an optimal balance between semantic loyalty and consistency.

### C.4. Video Editing on HunyuanVideo

Figure 17 illustrates the results on complex motion sequences, where T-Edit successfully executes attribute mod-ifications and style transfers while preserving the original motion trajectories and fine-grained temporal details.

## D. Future Work

This work analyzes and leverages the dynamic distribution of structural information across both layer and timestep dimensions in MMDiTs architectures through T-Edit, providing a structure-aware modeling perspective for generative image editing. Future work may further explore the intrinsic structural control capabilities of MMDiTs, for example by introducing spectral constraints during pretraining to promote more effective decoupling between semantic and geometric representations. In addition, extending the framework to maintain consistency in ultra-long video sequences and investigating modality-agnostic patterns of structural energy distribution remain promising research directions. We hope that these observations can contribute to a deeper understanding of internal representation mechanisms in large-scale Transformers and inspire the development of more robust multimodal generative systems.

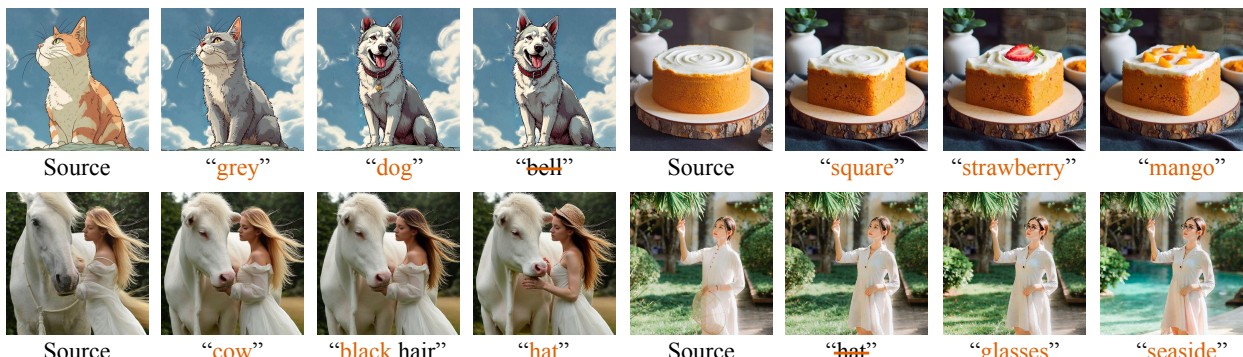

Figure 14. **Additional single-step image editing results with baseline comparisons.** T-Edit consistently preserves structural layout while enabling precise semantic edits under diverse prompts.

Figure 15. **Additional multi-step editing results.** T-Edit maintains structural consistency across sequential prompt modifications, demonstrating stable compositional editing behavior.

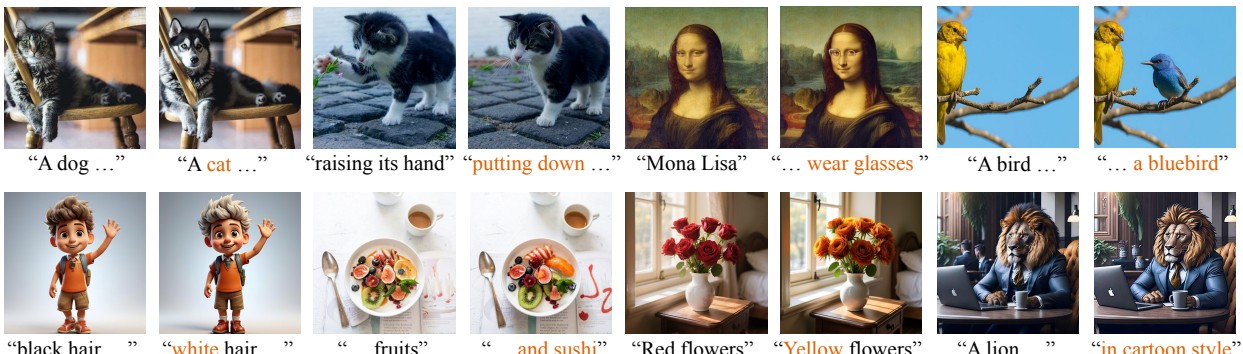

"A dog …"  "A cat …"  "raising its hand"  "putting down …"  "Mona Lisa"  "… wear glasses"  "A bird …"  "… a bluebird"

"black hair …"  "white hair …"  "… fruits"  "… and sushi"  "Red flowers"  "Yellow flowers"  "A lion …"  "in cartoon style"

*Figure 16.* **Editing results on Stable Diffusion 3.** T-Edit generalizes effectively to SD3, demonstrating architecture-agnostic applicability on MMDiTs-based diffusion models.

"A person is speaking." ⟶ "A person wearing gold-rimmed glasses is speaking."

"A stream in the wild." ⟶ "A stream in the wild in winter."

"A cat lowers its head." ⟶ "A dog lowers its head."

"A person is dancing." ⟶ "Iron Man is dancing."

"A girl is speaking." ⟶ "A boy is speaking."

"A flying passenger plane." ⟶ "A flying passenger plane in a cartoon style."

*Figure 17.* **Additional video editing results.** T-Edit preserves motion trajectories and temporal consistency while enabling appearance and attribute edits.

