# OpenReview forum: "T-Edit: Triple-Branch Diffusion Anchoring for Consistent Editing"
_ICML.cc/2026/Conference — ICML 2026 regular_

### Official Review · Reviewer_4orp · 2026-02-25

**Soundness:** 3
**Presentation:** 3
**Significance:** 3
**Originality:** 3
**Overall Recommendation:** 5
**Confidence:** 4

**Summary:**

This paper proposes T-Edit, a training-free image editing framework designed to address semantic distortion and structural inconsistency in Multi-modal Diffusion Transformers (MMDiTs). The authors formalize consistent editing as a trajectory anchoring process. By decoupling the pipeline into three branches—Inversion, Reconstruction, and Editing—the framework utilizes the reconstruction branch as a structural reference to compensate for latents manifold deviations in real-time.

Furthermore, the paper provides an in-depth analysis of the internal conditioning mechanisms of DiTs, revealing the spatio-temporal heterogeneity of layer-wise structural sensitivity. Consequently, a Dynamic Vital Layer (DVL) localization mechanism based on information energy is proposed. To address the asymmetry of text-induced perturbations in the frequency domain, the authors introduce a Frequency-aware Anchoring (TA) strategy based on tensor Singular Value Decomposition (t-SVD) to preserve high-rank structural components. Experimental results demonstrate that T-Edit achieves state-of-the-art (SOTA) performance in semantic alignment and structural fidelity across single-step, multi-step image editing, and video editing scenarios.

**Compliance With Llm Reviewing Policy:**

Affirmed.

**Final Justification:**

The authors address my concerns. I think it is a technically solid paper, and training-free editing methods are still necessary nowadays for its flexibility and adaptability. So I raised the score from 4 to 5 and believe that this paper meets the standard of ICML.

**Key Questions For Authors:**

Compared to single-branch methods (e.g., RF-Edit) or mask-based methods (e.g., KV-Edit), what is the specific increase in GPU memory footprint and wall-clock time for T-Edit’s triple-branch execution during a single inference?

The information retention rate is set to $\tau=0.9$. Is this parameter consistently robust across models of different scales (e.g., from Stable Diffusion 3 to FLUX.1)? Are there cases where model-specific tuning is required?

In the ablation experiments (Table 4), the "+ Error Correction" component shows a significant improvement in LPIPS. Could you clarify how this error correction signal is specifically weighted and injected into the editing branch? Furthermore, how sensitive is this signal to different Guidance Scales?

**Strengths And Weaknesses:**

Strengths

- Innovative Triple-Branch Architecture: By introducing a reconstruction branch to quantify and compensate for "cumulative structural drift," the method effectively mitigates semantic bleeding caused by the joint text-image updates inherent in DiT architectures.

- Dynamic Layer Selection Mechanism: Unlike prior heuristic methods with fixed layer selection, the proposed DVL mechanism dynamically identifies attention layers with the highest structural impact based on image content and diffusion stage, enhancing the model's adaptability and generalization.

- Frequency-Domain Analytical Perspective: The authors observe that editing perturbations are primarily concentrated in high-frequency attention components. The innovative use of t-SVD for frequency-aware anchoring provides a valuable perspective on the internal stability of DiTs.

- Extensive Applicability: As a training-free method, T-Edit seamlessly extends to multi-step and video editing tasks. It outperforms mainstream MMDiT editing methods on benchmarks such as PIE-Bench in terms of FID and CLIP scores.

Weaknesses

- Increased Computational Overhead: Although the SVD decomposition in DVL is relatively efficient, the parallel execution of three branches (Inversion, Reconstruction, and Editing) and step-wise frequency-aware computations inevitably increase inference latency and GPU memory consumption. A quantitative analysis of inference efficiency (e.g., wall-clock time comparisons) is currently insufficient.

- Simplified Treatment of Video Editing: For video tasks, the authors employ a first-frame anchoring strategy to prevent flickering. While simple and effective, this approach may fail to capture dynamic structural features in long videos or scenes with significant background changes.

- Limitations in Extreme Editing Scenarios: While performing well in object replacement and attribute modification, the anchoring mechanism might be overly conservative in non-rigid editing scenarios involving significant topological changes (e.g., changing a "standing person" to a "crouching person"), potentially suppressing necessary semantic transformations.

---

> ### Author Rebuttal · Authors · 2026-03-31
>
> We appreciate your recognition of the innovative mechanisms and broad applicability of this research. In response to your key questions, we provide the following detailed clarifications.
>
> D1: Inference Overhead
>
> While Appendix B presents a comparison of inference times, we provide a precise quantitative analysis here.
>
> (1) Inference Time: By employing the snapshot method to optimize Singular Value Decomposition (SVD), T-Edit's runtime at $512 \times 512$ resolution is 43.37 seconds. This is significantly superior to the optimization-based RF-Edit (65.57 seconds) and remains in the same tier as the mask-based KV-Edit (39.39 seconds).
>
> (2) GPU VRAM: Using FLUX.1-dev-FP16 as the base model, conventional generation occupies 22GB of VRAM. Upon introducing the T-Edit framework, peak VRAM increases to 27GB, representing a marginal increase of only about 5GB. This efficiency is achieved through an asynchronous CPU offloading strategy, where all intermediate latent states, layer indices, and K/V pairs are transferred to CPU memory immediately after generation.
>
> D2: Regarding the Robustness of Information Retention Rate $\tau$
>
> We found that for both FLUX.1 and Stable Diffusion 3, the information energy distribution across attention layers is extremely skewed, meaning a small number of layers dominate the process. We randomly sampled energy scores across 100 steps, as shown below:
>
> | Score Range | Proportion | Cumulative Proportion |
> | :--- | :--- | :--- |
> | [0.000, 0.005) | 89.12% | 89.12% |
> | [0.005, 0.015) | 2.46% | 91.58% |
> | [0.015, 0.050) | 1.40% | 92.98% |
> | [0.050, 0.150) | 0.70% | 93.68% |
> | [0.150, 0.200] | 6.32% | 100.00% |
>
> Consequently, a fixed $\tau = 0.9$ serves as a stable threshold. The experiments for SD3 in Appendix C.3 followed this same setting and successfully achieved high-fidelity consistency.
>
> D3: Error Correction and Guidance Scale Sensitivity
>
> (1) Injection Method of Correction Signals: At denoising timestep $t$, the deviation between the inversion trajectory $Z\_{t}$ and the reconstruction trajectory $Z\_{t}^{R}$ is calculated in real-time to quantify the structural drift induced by inversion: $\delta\_{t}^{drift} = Z\_{t} - Z\_{t}^{R}$. This vector acts as an additive compensation signal directly added to the editing branch, with the mathematical update form: $Z\_{t}^{E*} = Z\_{t}^{E} + \delta\_{t}^{drift}$.
>
> (2) Sensitivity to Guidance Scale: Appendix A.1 verifies the sensitivity of numerical drift to text prompts. By calculating the reconstruction error for the source prompt $\vec{e}\_{src}(t) = Z\_t - Z\_t^R(c)$ and the target prompt $\vec{e}\_{tgt}(t) = Z\_t - Z\_t^R(c')$, data indicates that in the first 10 denoising steps, the cosine similarity between the two consistently exceeds 0.8. This suggests that numerical drift in the early stages is largely independent of the prompts.
>
> When denoising reaches step 14, the semantic guidance of the prompt strengthens, and similarity decreases significantly. Because the direction of drift in early denoising is highly consistent and prompt-independent, the generated structure deviation vector remains stable regardless of how much the Classifier-Free Guidance (CFG) scale is increased. Thus, the correction signal is highly insensitive to variations in the conventional guidance scale and can stably offset numerical divergence without interfering with mid-to-late stage semantic generation.
>
> D4: Limitations and Extreme Cases
>
> (1) In non-rigid editing involving large-scale topological deformations (e.g., changing "standing" to "squatting"), the reconstruction branch indeed acts conservatively as a structural regularizer; this is an explicit trade-off made for consistency. While the first-frame anchoring strategy in video tasks is efficient and stable for short sequences, it may struggle to capture evolving structures in long videos. We intend to explore the introduction of sliding-window re-anchoring to effectively mitigate temporal tension in extended video sequences.
>
> (2) To further demonstrate the boundaries of our method, we showcase more extreme cases on our anonymous homepage (https://anonymous.4open.science/r/T-Edit-5D6B). In cases involving large object replacement (Case 1) and non-rigid editing (Case 2), T-Edit precisely achieves semantic transitions while maintaining background and subject structural consistency. For localized edits (Case 3), the model responds accurately while preserving high subject identity consistency. When target prompt semantics conflict strongly with the source image (Case 4), the model tends to maintain the structure conservatively. This is also objectively limited by the data distribution of the pre-trained model—conflicting semantics exceed the inherent manifold. This reflects the trade-off for consistency and the generalization boundaries of the foundational model.

---

> > ### Author Rebuttal · Reviewer_4orp · 2026-04-01
> >
> > Thanks for the authors' response. After carefully reading the rebuttal and reviews by other reviewers, most of my concerns are addressed.
> >
> > In general, I think this is a technically solid paper with satisfying performance. Its training-free nature makes it easy to apply to various RF-based models, including different architectures and modalities. After further consideration, I would like to increase my rating.

---

> > > ### Author Response · Authors · 2026-04-06
> > >
> > > Thanks for your acknowledgment of our work and for raising the score. Thanks again for your time and effort in reviewing our paper.

---

### Official Review · Reviewer_uGsY · 2026-03-02

**Soundness:** 2
**Presentation:** 2
**Significance:** 1
**Originality:** 2
**Overall Recommendation:** 2
**Confidence:** 5

**Summary:**

The paper proposes T-Edit, a training-free framework decoupling inversion, reconstruction and editing trajectories to anchor structureal consistency dynamically. DVL and t-SVD anchoring are introduced to identify structure-critical vital attention layers and suppress high-rank semantic perturbations in non-vital layers to preserve fine-grained details. Experiments demonstrate the effectiveness of T-Edit against previous training-free methods.

**Compliance With Llm Reviewing Policy:**

Affirmed.

**Final Justification:**

This training-free method  still lags significantly behind the performance of open-sourced Qwen-Image Edit (released last year ) on standard image editing benchmark.  For this reason, I regret to say that I do not believe this paper meets the acceptance standard for ICML.

**Key Questions For Authors:**

What is the advantage of T-Edit over the popular open-sourced models Qwen-Image-Edit?

**Limitations:**

Not discussed

**Strengths And Weaknesses:**

Strength:

1. The paper is clearly written and well-organized.

2. The proposed method T-Edit could be applied to both image and video domain.

3. The experiments seem concrete and comprehensive. The evaluation against previous training-free methods demonstrate its effectiveness.

Weakness:

1. No user study is provided. Only statistical results which are usually not convincing in AIGC domain.

2. T-Edit is a training-free method, the performance is still far from models like Qwen-Image-Edit etc. From my perspective, training-free is a dead end for image editing.

3. T-Edit requires specific instruction pairs to conduct editing, which is problematic in real-world scenarios. Instead, image editing models like Qwen Image Edit supports free-form prompt.

4. The paper only shows image editing results with one image reference. Could it conduct multi-reference image editing/customization which is supported by many recent image editing models?

---

> ### Author Rebuttal · Authors · 2026-03-31
>
> We appreciate your recognition of the clarity of the paper, the cross-domain applicability of our method, and the comprehensiveness of our experiments. In response to your questions, we provide the following clarifications.
>
> C1: User Study
>
> We conducted a user study to evaluate the image editing quality of T-Edit. A total of 20 participants were invited to complete 20 sets of pairwise comparisons, evaluating three dimensions: Text Alignment, Structure Fidelity, and Overall Visual Quality. A total of 400 data points were collected, as detailed below (values represent T-Edit's win rate against each baseline):
>
> | Baseline Method | Text Alignment | Structure Fidelity | Overall Quality |
> | :-------------- | :------------- | :----------------- | :-------------- |
> | P2P             | 94.44%         | 90.45%             | 91.11%          |
> | MasaCtrl        | 92.45%         | 88.18%             | 90.91%          |
> | Flow-Edit       | 80.21%         | 74.55%             | 79.65%          |
> | RF-Edit         | 85.00%         | 77.06%             | 74.26%          |
> | KV-Edit         | 84.29%         | 71.43%             | 73.91%          |
> | FreeFlux        | 80.55%         | 80.43%             | 81.25%          |
> | Stable Flow     | 83.31%         | 81.46%             | 84.32%          |
>
> T-Edit significantly outperforms all baseline methods across all evaluation dimensions. In terms of Text Alignment, T-Edit achieved a win rate exceeding 80% in all cases, demonstrating its substantial advantage in prompt loyalty. Regarding Structure Fidelity, T-Edit maintained a 71.43% win rate even against the mask-based KV-Edit. In Overall Visual Quality, T-Edit is significantly superior to other comparison methods also based on the FLUX.1 model. These experimental results strongly support the superiority of T-Edit in practical applications.
>
> C2: Comparison with Qwen-Image-Edit
>
> On our anonymous project page (https://anonymous.4open.science/r/T-Edit-5D6B), we have added intuitive visual comparisons between T-Edit and the official Qwen-Image-Edit across various scenarios.
>
> (1) As a model trained on massive datasets, Qwen-Image-Edit performs exceptionally well in terms of generative freedom and local texture quality. However, it exhibits instability in maintaining structural consistency: for example, in the large-scale object replacement (horse) in Case 1, Qwen significantly altered the background and the person's pose; in the non-rigid editing of Case 2, uncontrolled expansion of the frame occurred; in Case 4, when the target semantics conflicted strongly with the source image (e.g., a "square-shaped head"), Qwen completely lost the underlying information of the reference image. In contrast, T-Edit demonstrates precision comparable to Qwen in the highly localized editing of Case 3, while providing extremely stable and high-fidelity structural consistency across all tests.
>
> (2) Qwen-Image-Edit has a high computational threshold; during local deployment, its VRAM usage reached approximately 60GB, and a single inference ($512 \times 512$) took about 2 minutes. The original intent of T-Edit was to deeply analyze and resolve the underlying mechanisms of structural drift in pre-trained MMDiTs during editing. It provides a highly transparent solution with only marginal computational overhead and can seamlessly generalize to other MMDiT models. This irreplaceable value in mechanistic exploration and cross-architectural generalization constitutes the unique academic significance of the training-free paradigm.
>
> C3: Instruction Input Methods and Research Boundaries
>
> (1) Regarding input methods, T-Edit requires complete descriptive prompts but does not rely on specific instruction pairs. It supports source/target prompt pairs or target prompts alone.
>
> (2) Regarding multi-image reference customization, the current work focuses on accumulated drift and semantic leakage caused by the MMDiT architecture in single-image zero-shot editing. Personalized customization with multiple reference images typically requires external injection and weight fine-tuning. Your suggestion is highly insightful. In future work, we look forward to introducing the dynamic layer selection and frequency-domain anchoring mechanisms proposed in this study into multi-reference image generation frameworks to address issues such as identity feature consistency attenuation in more complex scenarios.

---

> > ### Author Rebuttal · Reviewer_uGsY · 2026-04-02
> >
> > I appreciate the authors' rebuttal, but from my perspective, it is not very persuasive.
> >
> > - For the qualitative comparison between Qwen-Image-Edit and T-Edit, it seems to me that Qwen-Image-Edit is better, especially for the horse replacing case.
> >
> > -  In addition, the authors did not provide quantitative comparison to the SoTA image editing models like StepFun Edit and Qwen-Image-Edit-2509.
> >
> > - No user study comparing Qwen-Image-Edit and T-Edit.
> >
> > -  Besides, it is a common case nowadays to compare image editing performance on large scale benchmarks, for example, [GEdit-Bench]( https://github.com/stepfun-ai/Step1X-Edit/tree/main/GEdit-Bench).

---

> > > ### Author Response · Authors · 2026-04-06
> > >
> > > Thank you for your continued attention and discussion. We would like to further clarify the fundamental differences between the training-free editing framework and large-scale instruction-tuned models.
> > >
> > > 1. Regarding the qualitative dispute over visual effects in the 'horse replacement' case, we acknowledge that large-scale fine-tuned models like Qwen-Image-Edit, leveraging their strong generative priors, can often produce visually more harmonious images. However, the core objective of this research is strictly defined as achieving precise editing with high fidelity and structural consistency, which fundamentally differs from the intrinsic logic of large end-to-end black-box models that tend to perform unconstrained redrawing. In an academic context where structural control is the primary consideration, alterations to non-edited regions (such as the subject's posture or background context) should be treated as **structural leakage** rather than as a successful controlled edit.
> > >
> > > 2. Regarding the question of not conducting quantitative comparisons with industrial-grade models like StepFun Edit or Qwen-Image-Edit-2509 on large-scale benchmarks such as GEdit-Bench, we sincerely point out that this comparison involves a severe **asymmetry** at the task definition level. Large-scale benchmarks like GEdit-Bench are specifically designed to evaluate a model's instruction-following capability, and their underlying logic naturally favors end-to-end models fine-tuned on massive instruction-image pairs at tremendous computational cost. In contrast, T-Edit is a zero-shot, target-prompt-driven training-free framework whose fundamental purpose is to reveal and control the internal evolution mechanisms of pre-trained MMDiTs.
> > > Forcing a training-free method into an instruction-based evaluation system requires converting editing instructions into target prompts. This transformation process inevitably introduces semantic degradation, failing to yield scientifically equitable quantitative conclusions. The baseline methods we selected, such as KV-Edit and FreeFlux, represent the most rigorous and applicable evaluation benchmarks currently available within the specific subfield of training-free diffusion editing.
> > >
> > > The academic contribution of T-Edit lies in transparently elucidating and suppressing the structural drift phenomenon within the MMDiTs architecture at a very low computational overhead, providing a mechanistic solution with cross-architecture generalization capabilities. We hope the above clarifications effectively address your concerns, and we once again thank you for your valuable time and professional review of our manuscript.

---

### Official Review · Reviewer_mbEq · 2026-03-10

**Soundness:** 3
**Presentation:** 3
**Significance:** 3
**Originality:** 3
**Overall Recommendation:** 4
**Confidence:** 3

**Summary:**

This paper studies the problem of structural drift and semantic leakage in text-driven image editing under MMDiT/DiT architectures. The authors argue that, unlike the more localized cross-attention editing behavior in U-Net-based models, MMDiT couples image and text tokens within a unified joint attention stream, so prompt perturbations propagate throughout the denoising trajectory and accumulate across multi-step edits, leading to background distortion, structure corruption, and identity inconsistency. To address this issue, the paper proposes T-Edit, a training-free triple-branch framework consisting of an inversion branch, a reconstruction branch, and an editing branch: the first records a reference trajectory, the second estimates and corrects editing drift, and the third performs anchored editing under the new prompt.

**Compliance With Llm Reviewing Policy:**

Affirmed.

**Key Questions For Authors:**

- The paper argues that harmful prompt-induced perturbations are mainly concentrated in high-frequency components. Does this finding hold consistently across different editing categories?
- Whether this method is applicable to more advanced editing frameworks, such as Qwen-Image-Edit ?

**Limitations:**

The main limitation of this paper is that the method appears empirically effective, but the mechanistic argument is not yet fully closed. The intuitions behind DVL and TA are reasonable, but the current evidence shows more that these designs work than that they are necessary, most natural, or broadly valid. The jump from empirical regularities to general conclusions still feels too large.

**Strengths And Weaknesses:**

Strengths:

- The paper is generally well written and easy to follow.
- The motivation is well articulated and credible.
- The empirical results are overall strong.
From the reported tables, T-Edit performs particularly well on semantic-quality metrics such as FID and CLIP, and also shows good stability in multi-step and video editing, suggesting that the method is indeed effective under the evaluation protocol used by the authors

Weaknesses:
- The cost analysis is insufficient.
- The generalization evidence remains limited. The main experiments are concentrated on FLUX.1 and Hunyuan. Although the paper claims that the method extends to different MMDiT architectures

---

> ### Author Rebuttal · Authors · 2026-03-31
>
> We thank the reviewers for their affirmation of the research motivation, writing logic, and superior performance of this paper. We provide detailed responses to your questions below.
>
> B1: Inference Cost Analysis
>
> Appendix B of the initial submission presented a comparison of average inference times at different resolutions; here, we further refine the cost analysis.
>
> (1) Peak GPU VRAM: When using FLUX.1-dev-FP16 as the base model, the conventional generation process occupies 22GB of VRAM. Upon introducing the T-Edit framework, the system's peak VRAM increases to 27GB. This 5GB increment demonstrates that the additional VRAM overhead introduced by T-Edit is marginal, and its overall VRAM ceiling is strictly limited by the underlying MMDiT backbone network. To achieve this memory efficiency, we employ an asynchronous offloading strategy, where all intermediate latent states, layer indices, and K/V pairs are transferred to CPU memory after generation, effectively alleviating VRAM pressure.
>
> (2) CPU Storage Overhead and Offloading Costs: T-Edit performs feature intervention only during the first 10 steps of the total 30-step denoising process. At a resolution of $512 \times 512$, with approximately 1GB of K/V storage overhead per step and less than 1GB for intermediate latent states and layer indices, the total CPU storage requirement is maintained at around 10GB. Furthermore, the transmission latency from GPU to CPU is only approximately 0.9 seconds, ensuring that cross-device data throughput does not constitute a significant performance bottleneck for overall inference efficiency.
>
> B2: Method Extensibility
>
> This study is not limited solely to FLUX.1 and HunyuanVideo when exploring architectural universality. We have demonstrated the seamless transfer and editing results of T-Edit on SD3 in Appendix C of the paper.
>
> As a representative model of the MMDiT architecture, SD3 adopts a dual-stream architecture for image and text embeddings, which is similar to the dual-stream blocks in FLUX.1. During migration, we did not need to reconstruct the algorithm; instead, we directly mapped the dynamic key-layer localization and frequency-domain anchoring mechanisms into the attention computation flow of SD3, successfully achieving high-fidelity consistent editing. This result proves that our method does not rely on empirical fitting to a single model but effectively captures the common patterns of information distribution in MMDiTs.
>
> B3: Prompt Perturbation Analysis
>
> While Figure 5 in the main text displays the frequency-domain distribution of prompt perturbations based on the average trend of the entire dataset, we here provide a supplementary comparative distribution categorized by task type.
>
> | Task Category | $\Delta_{High}$ | $\Delta_{Low}$ | $\Delta_{High}/\Delta_{Low}$ |
> | :--- | :--- | :--- | :--- |
> | Add Object | 845.2 | 541.8 | 1.56 |
> | Remove Object | 820.6 | 550.7 | 1.49 |
> | Change Object | 860.1 | 535.4 | 1.60 |
> | Change Posture | 815.3 | 562.2 | 1.45 |
>
> From the quantitative data across different editing task subsets, it can be observed that the high-frequency component differences (L2 norm) induced by target prompt perturbations significantly exceed the low-frequency component differences. This result strongly demonstrates that manifold deviations induced by prompts dominate in local high-frequency features, further consolidating the applicability of the frequency-aware anchoring mechanism across various editing scenarios.
>
> B4: Qwen-Image-Edit Applicability
>
> The original intent of T-Edit was to explore the inherent zero-shot editing potential of pre-trained MMDiTs, whereas Qwen-Image-Edit belongs to the category of editing models trained on large-scale image-text pairs. Nevertheless, the patterns of hierarchical structural sensitivity and frequency-domain asymmetry revealed by T-Edit can certainly be introduced as regularization priors into fine-tuning frameworks. For example, the DVL mechanism could be used to dynamically constrain gradient update layers during fine-tuning, or a frequency penalty term could be introduced into the training loss, thereby enhancing the stability of advanced instruction-based editing frameworks when dealing with complex structures.
>
> B5: Limitation Analysis
>
> The parsing of internal deep causal relationships within MMDiTs remains an open problem; this research has endeavored to provide rigorous empirical support. For instance, in Figure 4, the dynamic information energy and DINOv2 structural loss exhibit a Pearson correlation coefficient as high as 0.93, providing a strong proxy metric for DVL. Furthermore, the ablation experiments in Table 4, through strict control of variables, prove that the DVL and TA modules are indispensable for suppressing structural collapse. As envisioned in the future work section, we will attempt to introduce dynamic key layers and spectral constraints during training to achieve complete decoupling of semantic and geometric representations.

---

> > ### Author Rebuttal · Reviewer_mbEq · 2026-04-02
> >
> > i keep my score at 4.

---

> > > ### Author Response · Authors · 2026-04-06
> > >
> > > We sincerely thank the reviewer for the time spent re-evaluating our manuscript. We noticed that the status was marked as "Partially resolved" with an indication of "follow-up questions." However, as no specific questions were listed in the comments, we would deeply appreciate it if the reviewer could provide further details or clarify the remaining concerns. We are fully committed to addressing any outstanding issues to improve the quality of this work.

---

### Official Review · Reviewer_hbZq · 2026-03-12

**Soundness:** 3
**Presentation:** 3
**Significance:** 2
**Originality:** 2
**Overall Recommendation:** 4
**Confidence:** 4

**Summary:**

This paper studies training-free text-guided editing for MMDiT-based diffusion models and identifies two main failure modes in this setting: accumulated structural drift during inversion/denoising and semantic bleeding caused by joint text-image attention. To address this, the paper proposes T-Edit, a triple-branch framework with inversion, reconstruction, and editing branches, where the reconstruction branch provides online correction for the editing trajectory. The method further introduces Dynamic Vital Layer localization to adaptively select structurally important layers over timesteps, and a frequency-aware t-SVD Anchoring module to inject structural guidance into non-vital layers. Experiments on PIE-Bench, multi-step editing, and a video editing extension suggest that the method achieves a strong balance between semantic alignment and structural preservation, and the ablations support the contribution of the main components.

**Compliance With Llm Reviewing Policy:**

Affirmed.

**Final Justification:**

The authors addressed my concern, so I remain positive about the paper and keep my score at 4.

**Key Questions For Authors:**

- Could the authors report peak GPU memory, CPU offloading cost, and storage overhead for per-step K/V caching in addition to runtime? This would clarify the practical tradeoff and improve reproducibility.
- Could the authors add a more explicit failure analysis, especially for large geometric edits, highly localized edits, or cases where semantic fidelity conflicts strongly with structural preservation? This would improve my confidence in the robustness claims.

**Limitations:**

Yes.

**Strengths And Weaknesses:**

Strengths

- The paper is technically coherent and the method is clearly motivated by the specific failure modes of MMDiT editing. The three-branch design, online error correction, dynamic layer selection, and TA module fit together into a consistent pipeline.
- The empirical section is fairly strong. On PIE-Bench, T-Edit achieves the best FID and CLIP among the listed baselines, while remaining competitive on LPIPS and PSNR. The paper also includes multi-step editing results, a video extension, and component ablations.
- While many ingredients are related to existing ideas, the paper offers a reasonably creative integration tailored to MMDiT editing, especially in how it combines online correction, adaptive vital-layer localization, and frequency-aware anchoring within one unified framework.

Weaknesses

- he method is training-free but still relatively heavy in practice because it relies on inversion, multiple branches, and per-step feature storage. The runtime is comparable to several inversion-based baselines, but still substantial, and peak memory tradeoffs are not fully quantified.

---

> ### Author Rebuttal · Authors · 2026-03-31
>
> We appreciate your affirmation of the mechanism-driven nature, methodological innovation, and empirical aspects of this research. We provide detailed responses to your concerns below.
>
> A1: Inference Cost Analysis
>
> To clarify the inference overhead of T-Edit, we have conducted a precise quantitative analysis of the model's resource consumption.
>
> (1) Peak GPU VRAM: Using FLUX.1-dev-FP16 as the base model, the generation process occupies 22GB of VRAM. Upon fully integrating the T-Edit framework, the system's peak VRAM increases to 27GB. This 5GB increment demonstrates that the memory overhead introduced by T-Edit is marginal, and the overall VRAM ceiling is strictly limited by the underlying MMDiT backbone network. To achieve this memory efficiency, we employ an asynchronous offloading strategy, where all intermediate latent states, dynamic key-layer indices, and attention Key/Value (K/V) pairs are immediately transferred to CPU memory after generation, thereby effectively reducing the VRAM burden. In addition, enabling `cpu_offload` can further reduce the peak video memory requirement to 20 GB to accommodate consumer-grade GPUs.
>
> (2) CPU Storage Overhead and Offloading Costs: Based on temporal analysis, T-Edit performs feature intervention only during the first 10 steps of the 30-step denoising process. At a resolution of $512 \times 512$, with approximately 1GB of K/V storage overhead per step and less than 1GB for intermediate latent states and layer indices, the total CPU storage requirement is maintained at around 10GB. Furthermore, the data transfer latency from GPU to CPU is only about 0.9 seconds, ensuring that cross-device data throughput does not constitute a significant performance bottleneck for overall inference efficiency. The detailed quantitative data regarding resource consumption mentioned above will be fully incorporated into the final version to maximize the engineering transparency and reproducibility of this research.
>
> A2: Extreme Case Analysis
>
> We present more extreme cases on our anonymous project page (https://anonymous.4open.science/r/T-Edit-5D6B).
> In cases involving large-scale object replacement (Case 1) and non-rigid editing (Case 2), T-Edit demonstrates excellent robustness, accurately achieving semantic transitions while maintaining high-fidelity topological consistency of the background and subject structures. For highly localized editing requirements (Case 3), the model also precisely responds to local edits while preserving high subject identity consistency.
>
> When target prompts conflict strongly with the source image (Case 4), T-Edit's performance is constrained. On one hand, when the target text attempts to induce drastic and counter-intuitive topological deformations, the information-energy-based reconstruction branch acts as a powerful structural regularizer, forcing the model to conservatively maintain the source image's structure. On the other hand, this is objectively limited by the data distribution of the pre-trained large model itself—attempting to generate geometries that completely deviate from realistic physical priors often exceeds the inherent semantic manifold of the base model. This represents a trade-off made for consistency and faithfully reflects the generalization boundaries of the foundational model.

---

> > ### Author Rebuttal · Reviewer_hbZq · 2026-04-01
> >
> > The authors addressed my concern, so I remain positive about the paper and keep my score at 4.

---

> > > ### Author Response · Authors · 2026-04-06
> > >
> > > Thank you for your recognition and positive response, as well as your assistance in improving our manuscript.

---

### Decision · Program_Chairs · 2026-04-30

**Decision:**

Accept (regular)

**Comment:**

The paper proposes T-Edit, a training-free consistent image editing method for diffusion models. It leverages the reconstruction branch as a structural reference, a DVL-based localization mechanism, and t-SVD to anchor high-rank structural components. Reviewers hbZq and uGsY recognized the thorough evaluation across diverse benchmarks. Reviewer 4orp highlighted the paper’s technical contributions compared with prior methods. Reviewer uGsY also acknowledged the paper’s writing quality and its flexibility for both image and video editing (uGsY , 4orp).

At the same time, the reviewers raised several concerns: (1) time efficiency and computational cost (hbZq, 4orp), (2) insufficient experiments and explanations (hbZq, mbEq, 4orp), as well as the absence of a user study (uGsY), (3) the applicability of the training-free setting and its performance gap compared with training-based methods such as Qwen-Image-Edit (uGsY), and (4) the simplified treatment of video editing and limitations in extreme editing scenarios (4orp).

During the rebuttal, the authors provided additional experimental results, a user study comparing against other training-free editing methods, and more detailed responses to the reviewers’ technical questions. However, reviewer uGsY noted that the user study still did not include some strong training-based baselines, such as StepFun Edit and Qwen-Image-Edit, and that the method was not evaluated on GEdit-Bench. As a result, uGsY remained concerned about the effectiveness and broader applicability of the proposed training-free method compared to training-based methods. During reviewer discussion, reviewer mbEq agreed with uGsY’s concern and changed their stance from weak accept to reject in the comment only visible to AC, but the overall recommendation score was not updated along with the final justification. In contrast, reviewer 4orp raised their score from weak accept to accept.

The final outcome was two rejects, one weak accept, and one accept. Although the AC agreed that including comparisons with Qwen-Image-Edit in the user study would have made the evaluation more comprehensive, the AC also recognized the value of advancing a training-free method with strong flexibility and adaptability. Taking these factors into account, the paper is recommended for weak acceptance.

Please also fix the following references and make the format of all references consistent:
No conference venue
1. Rombach, R., Blattmann, A., Lorenz, D., Esser, P., and Ommer, B. High-resolution image synthesis with latent
diffusion models, 2021.
2. Kirstain, Y., Polyak, A., Singer, U., Matiana, S., Penna, J., and Levy, O. Pick-a-pic: An open dataset of user preferences for text-to-image generation. 2023.
3.  Meiri, B., Samuel, D., Darshan, N., Chechik, G., Avidan, S.,
and Ben-Ari, R. Fixed-point inversion for text-to-image diffusion models. CoRR, 2023